# Community-Level Sports Group Participation and Health Behaviors Among Older Non-Participants in a Sports Group: A Multilevel Cross-Sectional Study

**DOI:** 10.3390/ijerph18020531

**Published:** 2021-01-10

**Authors:** Taishi Tsuji, Satoru Kanamori, Yasuhiro Miyaguni, Katsunori Kondo

**Affiliations:** 1Faculty of Health and Sport Sciences, University of Tsukuba, Bunkyo City, Tokyo 112-0012, Japan; 2Graduate School of Public Health, Teikyo University, Itabashi City, Tokyo 173-8605, Japan; satoru_kanamori@med.teikyo-u.ac.jp; 3Department of Preventive Medicine and Public Health, Tokyo Medical University, Shinjuku City, Tokyo 160-8402, Japan; 4Center for Gerontology and Social Science, National Center for Geriatrics and Gerontology, Obu City, Aichi 474-8511, Japan; y.miyaguni@ncgg.go.jp (Y.M.); kkondo@chiba-u.jp (K.K.); 5Center for Preventive Medical Sciences, Chiba University, Chiba City, Chiba 263-8522, Japan

**Keywords:** multilevel analysis, social capital, contextual effect, housebound, exercise epidemiology

## Abstract

This study validates the relationship between community-level sports group participation and the frequency of leaving the house and transtheoretical model stages of behavior change for exercise among older individuals who did not participate in a sports group. We used cross-sectional data from the 2016 Japan Gerontological Evaluation Study. The proportion of sports group participants at the community level was calculated using the data from 157,233 older individuals living in 1000 communities. We conducted a multilevel regression analysis to examine the relationship between the proportion of sports group participants and the frequency of leaving the house (1 day/week or less) and the transtheoretical model stages of behavior change for exercise. A statistically significant relationship was observed between a high prevalence of sports group participation and lower risk of homeboundness (odds ratio: 0.94) and high transtheoretical model stages (partial regression coefficient: 0.06) as estimated by 10 percentage points of participation proportion. Older individuals, even those not participating in a sports group, living in a community with a high prevalence of sports group participation are less likely to be homebound; they are highly interested and have numerous opportunities to engage in exercise.

## 1. Introduction

Older individuals engaging in sports and exercises while participating in groups are at lower risk of functional disability, depressive symptoms, and falls than those engaging in sports and exercises alone without participating in such groups [1,2,3]. The suspected mechanisms underlying health outcomes that are more likely to be obtained through sports group participation rather than individual exercise involve the benefits of physical (e.g., inducing good adherence and long duration of exercise) [4,5,6], psychological (e.g., enjoyment, enhanced self-esteem, and decreased stress) [4,7], and social (e.g., receiving social support, social capital, and social influence) factors [4,7,8]. Furthermore, community-level sports group participation had a preventive effect on depressive symptoms [9] and cognitive impairment [10] in older individuals. This result indicated that regardless of participation in a sports group, older individuals living in communities with a high prevalence of sports group participation were less likely to present with depressive symptoms and develop cognitive impairment than those living in a community with a low prevalence of participation. However, the reason why older individuals who do not participate in community sports groups remain healthy is not fully elucidated.

Civic participation, such as sports group participation, is a component of social capital [11]. One of the pathways between community-level social capital and individual-level health outcomes is social contagion, referring to the notion that behaviors spread more quickly through a tightly knit social network [12]. The social contagion extends to older individuals who may be encouraged by sports group participants to increase their interest and prepare to engage in sports and exercise activities. As a result, even if individuals do not join a sports group, several older individuals may frequently go out, have an interest in engaging in exercise and sports, and acquire those habits. A decrease in the frequency of going out and being homebound are known markers of complex comorbidities and vulnerability, and a specific population who is at risk possesses these characteristics [13]. Furthermore, in reference to the transtheoretical model stages of behavior change (i.e., precontemplation, contemplation, preparation, action, and maintenance stages), staying at a higher stage is considered a healthy lifestyle [14]. From the public health and health promotion perspectives, establishing an effective approach to progress the stage of behavior change is required [15].

This multilevel cross-sectional study examines whether the frequency of leaving the house was maintained and the transtheoretical model stages of behavior change for exercise were high among older individuals, even those who do not participate in the sports group, living in a community with numerous sports group participants.

## 2. Materials and Methods

### 2.1. Study Design and Participants

We used cross-sectional data from the Japan Gerontological Evaluation Study (JAGES), which is an ongoing cohort study exploring social, environmental, and behavioral factors correlated with health loss, particularly functional decline or cognitive impairment, among individuals aged 65 years or older [16]. We mailed a set of questionnaires to 279,661 community-dwelling individuals between October 2016 and January 2017, and these individuals were randomly selected from 39 municipalities, including metropolitan, urban and semi-urban, and rural communities, in 18 prefectures from as far north as Hokkaido (i.e., the northernmost prefecture) and as far south as Kyushu (i.e., the southernmost region) in Japan. The exclusion criteria were as follows: individuals receiving support and long-term care certification under the Japanese long-term care insurance system [17]; those having limitations in performing activities of daily living, which is defined as inability to walk, bathe, or use the toilet without assistance; and those living in communities with ≤30 respondents. One community was defined primarily by the school district. In the procedure of generating community-level variables of sports group participation, respondents who did not answer the extent of sports group participation were considered as “non-participants” in a sports group regardless of the status of individual exercise, and aggregated individual-level sports group participation according to community area was considered a community-level independent variable. With regard to the procedure of conducting the main statistical analysis, we further excluded respondents with incomplete information on sex, extent of sports group participation, or each outcome variable or those who participated in a sports group 1 day/month or more.

### 2.2. Measurements

#### 2.2.1. Frequency of Leaving the House

The participants were assessed for the frequency of leaving the house (including to one’s garden, immediate neighborhood, shopping complex, and hospital). The choices for the answers were as follows: ≥4 days/week, 2–3 days/week, 1 day/week, 1–3 days/month, a few times/year, and zero. We defined going out 1 day/week or less as mostly homebound [13,18], and this was previously associated with a higher mortality risk among community-dwelling older individuals [18].

#### 2.2.2. Transtheoretical Model Stages of Behavior Change for Exercise

The transtheoretical model stages of behavior change assess an individual’s readiness to act on a healthier behavior [14]. In this study, the participants were asked the question, “which of the following best describes your current lifestyle?” Here, regular exercise is defined as performing exercise once a week or more for at least 20 min per session. The choices for the answers were as follows: (1) not engaging in regular exercise and the lack of intention to start exercising in the future, (2) have not started exercising yet but committed to taking action within 6 months, (3) performing minimal exercises but not consistent, (4) exercising consistently for less than 6 months, and (5) exercising consistently for 6 months or more. We referred to the question item that was validated for Japanese adults [19]; however, we revised the frequency of regular exercise from twice a week to once a week, considering that the participants of this study were older individuals. The series had five stages, that is, precontemplation, contemplation, preparation, action, and maintenance, and an individual will go through these stages in adopting a healthy behavior or quitting an unhealthy behavior [14,20,21].

#### 2.2.3. Community-Level Sports Group Participation

The participants were assessed for the frequency of sports group participation. The choices for the answers were as follows: ≥4 days/week, 2–3 days/week, 1 day/week, 1–3 days/month, a few times/year, and zero. We defined participating 1 day/month or more as participation in a sports group [9,22], and aggregated individual-level sports group participation according to the community area defined primarily by the school district was considered a community-level independent variable. A study has indicated a strong correlation between the proportion of older individuals with poor self-rated health and depressive symptoms living in areas with community-level sports group participation once a month or more (r = −0.233 and −0.355, respectively) compared with community-level sports group participation once a week or more (r = −0.210 and −0.314, respectively) [22].

#### 2.2.4. Covariates

We evaluated confounding variables between community-level sports group participation and individual-level health status [9]. Data on basic demographic characteristics, including sex and age, were collected. The participants were divided into the following age groups: 65–69, 70–74, 75–79, 80–84, and ≥85 year groups. Then, they were queried on their household members and categorized as living with others or living alone. Drinking and smoking status (none, past, or current) and education (<10, 10–12, or ≥13 years) were classified according to each answer. Annual equivalent income was calculated by dividing the household income by the square root of the number of household members and categorized into the following three groups: JPY <2,000,000; JPY 2,000,000–3,999,999; and JPY ≥4,000,000. To obtain information about disease status in treatment, the participants were asked if they were currently receiving any medical treatment or had secondary diseases; the answer was either yes or no. If the participants did not respond to the individual-level covariates, corresponding observations were assigned to the missing categories. As community-level covariates, population density per km^2^ of inhabitable area and mean annual household income for each community area were calculated and categorized into the following quartile categories: ≥10,000; 7000–9999; 2500–6999; and <2500 individuals per km^2^ and JPY ≥2,713,000; JPY 2,468,000–2,712,999; JPY 2,250,010–2,467,999; and JPY <2,250,010, respectively.

### 2.3. Statistical Analysis

This analysis was conducted from January 2019 to March 2019. The multilevel analysis framework assumes that the health outcome of an individual partly depends on the community in which an individual lives. Multilevel models estimate the variations in outcome between communities (random effects) and the effects of community-level variables while adjusting for individual- and community-level characteristics (fixed effects). Multilevel logistic regression analysis was conducted to calculate the odds ratio (OR) and 95% confidence interval (CI) for being mostly homebound, and multilevel mixed-effects linear regression analysis was performed to calculate the partial regression coefficient (*B*) and 95% CI for the transtheoretical model stages of behavior change for exercise. The OR and *B* of community-level sports group participation was estimated as a 10-percentage point change in aggregated sports group participation. The following three models of analysis were used: the null model, crude model including community-level sports group participation, and fully adjusted model (the crude model + all covariates). We calculated the proportional changes in the variance of the crude and fully adjusted models to the null model to estimate how much the community-level variances of the frequency of leaving the house and transtheoretical model stages of change were explained by the exposure and covariates. Furthermore, we conducted a subgroup analysis by age groups (65–69, 70–79, and ≥80 years) to investigate whether the relationships were consistent or not. Analyses were performed using Stata/MP 14.2 (Stata Corp., College Station, TX, USA).

## 3. Results

Figure 1 shows the flow of the participants in this study. We received responses from 196,438 individuals, with a response rate of 70.2%. We excluded 38,707 respondents who received support and long-term care certification or who have limitations in performing activities of daily living and 498 respondents who lived in communities with ≤30 respondents. Thus, 157,233 respondents (mean and standard deviation for age: 73.8 ± 6.0 years) living in 1000 communities. They were considered analytic participants for generating community-level variables of sports group participation and mean annual household income. Furthermore, we excluded respondents with incomplete information on sex (*n* = 23) or extent of sports group participation (*n* = 25,641) or those who participated in a sports group 1 day/month or more (*n* = 40,879). Individuals with missing information on each outcome were excluded as well. Finally, 89,847 participants were considered analytic participants for the frequency of leaving the house analysis. With regard to the transtheoretical model stage analysis, the analytical sample comprised 10,487 participants because the question item was distributed to one-eighth of the participants who were randomly allocated.

Table 1 shows the demographic characteristics of the participants according to each analysis. Among the 89,847 analytic participants of the frequency of leaving the house analysis, 7364 (8.2%) reported going out 1 day/week or less and were categorized as mostly homebound. Among the 10,487 analytic participants for the transtheoretical model stage analysis, approximately 40% (*n* = 4356) were at the precontemplation stage, whereas 20% (*n* = 2260) were at the action or maintenance stage. Table 2 shows the descriptive statistics for the community-level variables. The mean proportion of sports group participants was 26.7% (standard deviation: 7.6%; range: 2.0–50.5%).

Table 3 and Appendix A show the results of the multilevel regression analyses. According to the analysis of the frequency of leaving the house, regardless of models including covariates, a higher prevalence of sports group participation is statistically significantly associated with a lower risk of homeboundness (OR: 0.94; 95% CI: 0.89–0.996 in the adjusted model), as estimated by 10-percentage points of participation proportion. The community-level variance decreased by 22.5% with the addition of the proportion of sports group participants and 36.3% with the addition of all covariates to the null model. In addition, community-level sports group participation was positively associated with the transtheoretical model stages of behavior change for exercise (*B*: 0.06; 95% CI: 0.01–0.12 in the adjusted model), and the community-level variance decreased by 23.1% and 38.5%, respectively. Appendix A shows the subgroup analysis by age groups. The direction of relationships was consistent among age groups although the 95% CIs were widened due to the smaller sample size.

## 4. Discussion

To the best of our knowledge, this is the first study to assess the contextual relationship between community-level prevalence of sports group participation and health behavior in older individuals, particularly those who do not participate in a sports group. A 10-percentage point increase in community-level sports group participation was associated with a 6% reduction in the risk of being homebound and a 0.06 higher in the transtheoretical model stages of behavior change for exercise after adjusting for potential confounders. These favorable associations can occur at any age group in the older population.

Social contagion might be considered a pathway that confirmed the results of this study supporting our hypothesis. Sometimes the behavior spreading via the network can promote healthy lifestyle changes (e.g., the spread of smoking cessation) [23,24]. In relation to this study, in communities where sports group participation is active, even if individuals do not participate in the group, they may have numerous opportunities to watch or cheer during the activities or participate in occasional events. As a result, they have numerous opportunities to go out, which may increase their interest in engaging in exercise and sports. Furthermore, Seino and colleagues have reported that community-level informal neighbor relationships were positively associated with individual-level moderate-to-vigorous physical activities among older men [25]. Both social participation, such as that in sports groups, and neighbor relationships are categorized as structural social capital; the affluence of this type of resource may enhance the frequency of going out and physical activities among older individuals living in the community.

Another possible pathway is collective efficacy, which is the group-level analog of the concept of self-efficacy and refers to the ability of the collective to mobilize to undertake a collective action [12,26,27]. Facilities, built environment, industries, systems, and bylaws for health promotion may develop to reflect the opinions and actions of communities with numerous sports groups and their participants. The group-level mechanisms of widespread sports group participation may result in positive spillover effects [9].

The transtheoretical model of behavior change is an integrative theory of therapy providing strategies or processes of change to guide an individual [14]. The strategies for enhancing the stages are mostly individual-oriented approaches, such as raising an individual’s consciousness and enhancing self-efficacy [28,29,30]. Naturally, the importance of controlling the environment is also mentioned; however, it is focused on the environment relatively close to an individual, which directly correlated with their consciousness and activities. A systematic review assessing the efficacy of dietary or physical activity interventions based on the transtheoretical model stages of behavior change in overweight and obese individuals has reported that the interventions led to sustained weight loss during the intervention period [15]. Another review has concluded that the transtheoretical model is a useful and suitable behavior model in creating, developing, and evaluating interventions to acquire and improve physical activity habits in older individuals [29]. However, evidence about whether such interventions could improve the stages of behavioral change and sustain them over time is limited. Meanwhile, the results of this study indicated that the stages may be improved by arranging a community environment where sports groups are active, even without working on older individuals directly. Assuming that there are 10-percentage points for more sports group participants in the community area, then 6 of 100 non-participants living in the area will be one stage higher. The accumulation of this clinical significance would not be a negligible contribution to public health.

The study strength is its large, nationwide, population-based sample enabling community- and individual-level multilevel analyses for clarifying the contextual relationship of sports group participation limited only to non-participants in a sports group. However, this study had several limitations. First, reverse causality could occur because of the nature of the cross-sectional design, and longitudinal studies must be conducted to resolve this limitation. Second, selection bias might have affected the results due to the relatively low response rate at 70.2%. According to our previous study, the response rate and percentage of sports group participants with older age were significantly lower than those with younger age [31]. Therefore, the participants in this study might be relatively at low risk of homeboundness and might stay at high transtheoretical model stages of behavior change for exercise, and those relationships might have been underestimated. Third, the frequency of leaving the house and the transtheoretical model stages of behavior change for exercise are assumed to be associated with the levels of physical activity/inactivity; however, we did not collect the information using a valid index. If we evaluate the levels of physical activity/inactivity, we could further define the pathway linking community-level sports group participation and health outcomes among older individuals living in the community. Fourth, we could not consider the geographic characteristics of the communities which could contribute to physical activity [32]. This might be one of the unmeasured confounders and diminish the relationships identified in this study.

## 5. Conclusions

Older individuals, even those who do not participate in a sports group, living in a community area with a high prevalence of sports group participation were less likely to be homebound and have higher transtheoretical model stages of behavior change for exercise than those living in an area with a low prevalence of participation. Promoting sports groups in a community may be an effective population-based strategy for increasing the frequency of leaving the house and enhancing interest in and providing opportunities to engage in exercise and sports even for non-participants in a sports group. A policy to increase sports groups, giving priority to communities where many older individuals with health problems live, may be effective.

## Figures and Tables

**Figure 1 ijerph-18-00531-f001:**
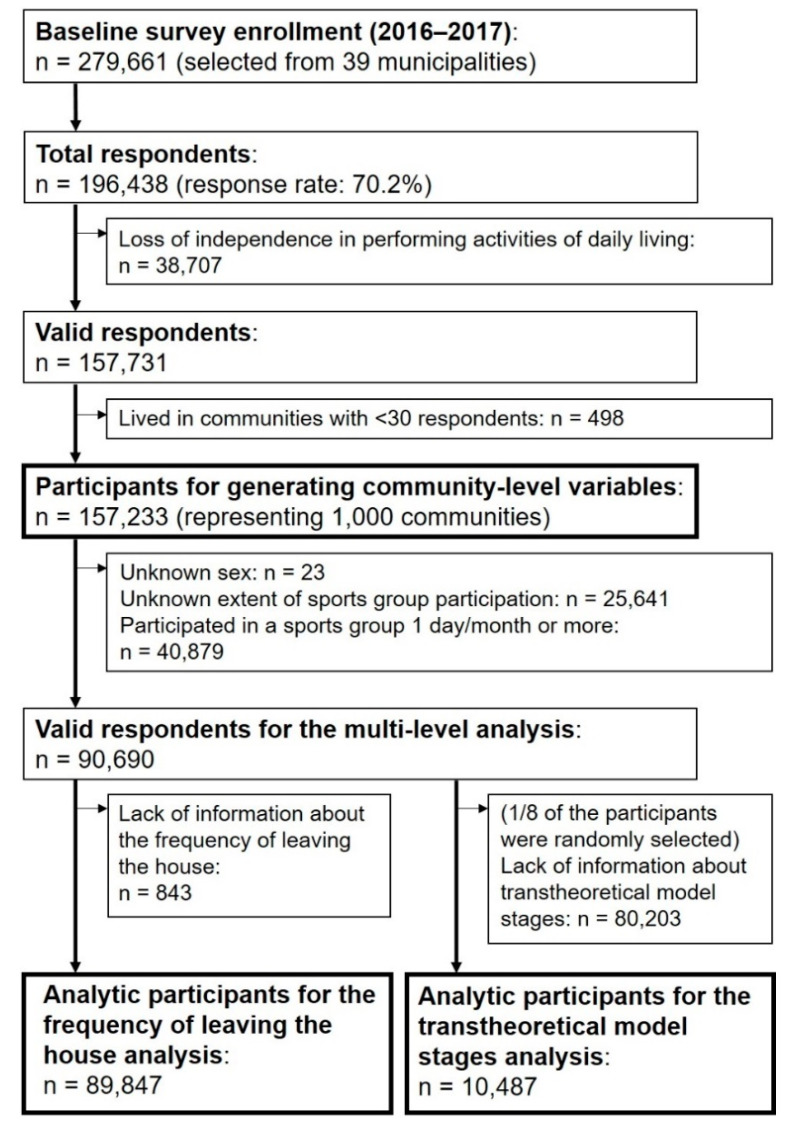
Flow of participants throughout the study.

**Table 1 ijerph-18-00531-t001:** Characteristics of the participants.

	Analytic Participants for the Frequency of Leaving the House Analysis		Analytic Participants for the Transtheoretical Model Stages Analysis	
	*n*	%	*n*	%
Total	89,847	100.0%	10,487	100.0%
Sex				
Men	45,847	51.0%	5356	51.1%
Women	44,000	49.0%	5131	48.9%
Age (years)				
65–69	30,924	34.4%	3713	35.4%
70–74	24,108	26.8%	2791	26.6%
75–79	18,821	20.9%	2166	20.7%
80–84	10,698	11.9%	1232	11.7%
≥85	5296	5.9%	585	5.6%
Disease status in treatment				
No	17,593	19.6%	2117	20.2%
Yes	69,631	77.5%	8093	77.2%
Missing	2623	2.9%	277	2.6%
Living with others				
Yes	76,657	85.3%	8957	85.4%
No (living alone)	9873	11.0%	1161	11.1%
Missing	3317	3.7%	369	3.5%
Drinking status				
None	43,819	48.8%	5062	48.3%
Past	10,228	11.4%	1201	11.5%
Current	34,504	38.4%	4066	38.8%
Missing	1296	1.4%	158	1.5%
Smoking status				
None	50,115	55.8%	5853	55.8%
Past	27,754	30.9%	3222	30.7%
Current	11,402	12.7%	1306	12.5%
Missing	576	0.6%	106	1.0%
Education (years)				
<10	29,024	32.3%	3230	30.8%
10–12	36,679	40.8%	4319	41.2%
≥13	23,142	25.8%	2831	27.0%
Missing	1002	1.1%	107	1.0%
Annual household equivalent income (Japanese Yen)				
<2,000,000	35,851	39.9%	4208	40.1%
2,000,000–3,999,999	28,200	31.4%	3327	31.7%
≥4,000,000	7970	8.9%	957	9.1%
Missing	17,826	19.8%	1995	19.0%
Frequency of leaving the house				
>1 day/week	82,483	91.8%		
≤1 day/week	7364	8.2%		
Transtheoretical model stages				
1. Precontemplation			4356	41.5%
2. Contemplation			1215	11.6%
3. Preparation			2822	26.9%
4. Action			365	3.5%
5. Maintenance			1729	16.5%
Mean (standard deviation)			2.4	(1.5)

**Table 2 ijerph-18-00531-t002:** Characteristics of community-level variables (*n* = 1000).

Proportion of Sports Group Participants (%)		
Mean (standard deviation)	26.7	(7.6)
(Minimum–Maximum)	(2.0–50.5)	
**Population Density** **(Persons Per km^2^ of Inhabitable Area), *n***		
≥10,000	257	
7000–9999	258	
2500–6999	238	
<2500	247	
**Mean Annual Household Equivalent Income** **(Japanese Yen), *n***		
≥2,713,000	249	
2,468,000–2,712,999	249	
2,250,010–2,467,999	249	
<2,250,010	249	
Missing	4	

**Table 3 ijerph-18-00531-t003:** Point estimates and 95% confidence intervals (CIs) estimated from multilevel regression analyses in each model.

	Null Model		Crude Model				Adjusted Model ^1^			
**Outcome: mostly homebound (leaving the house 1 day/week or less)**										
(number of communities = 1000, number of participants = 89,847)										
Fixed effects										
Proportion of sports group participants										
Per 10 percentage points, OR (95% CI)			0.82	(0.79–0.86)			0.94	(0.89–0.996)		
Random effects										
Community-level variance										
Ωμ, (standard error)	0.102	−0.012	0.079	−0.011			0.065	−0.011		
Proportional changes in variance, %			22.5				36.3			
**Outcome: transtheoretical model stages of behavior change for exercise**										
(number of communities = 988, number of participants = 10,487)										
Fixed effects										
Proportion of sports group participants										
Per 10 percentage points, *B* (95% CI)			0.09	(0.05–0.14)			0.06	(0.01–0.12)		
Random effects										
Community-level variance										
Ωμ, (standard error)	0.026	−0.009	0.02	−0.009			0.016	−0.008		
Proportional changes in variance, %			23.1				38.5			

OR: odds ratio, CI: confidence interval. ^1^ Adjusting for age, sex, population density, mean annual household equivalent income (community-level), disease status in treatment, living status, drinking status, smoking status, education, and annual household equivalent income (individual-level).

## Data Availability

Data are from the JAGES study. All enquiries are to be addressed at the data management committee via e-mail: dataadmin.ml@jages.net. All JAGES datasets have ethical or legal restrictions for public deposition due to inclusion of sensitive information from the human participants.

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
