# Peer review of "Community-Level Sports Group Participation and Health Behaviors Among Older Non-Participants in a Sports Group: A Multilevel Cross-Sectional Study"

_ijerph, 2021, doi:10.3390/ijerph18020531_

Round 1

Reviewer 1 Report

The homeboundness among elderly is a serious problem investigated in this study. The authors pointed the benefits of regular physical activity in different communities which is a good direction for public health priorities.

My main concern refers to the literature cited by the Authors. 8 out of 22 (36%!) positions are self-citations. They are relevant to the article, as they are mainly the papers already based on the same study (JAGES), however I would like the Authors to find more international literature to refer to. Worth noticing fact is that in the discussion section the Authors refer to only 5 papers published by other authors. I would encourage them to discuss more international literature, as the discussion section should contain more comparison and references to existing findings.

I would also advise to simplify the title of the paper as it can be confusing for the readers.

Author Response

Thank you for your time to review our manuscript and providing constructive feedback. We think that our manuscript is improved by addressing your feedback. Thank you very much.

We highlighted our changes using the Track Changes function in the revised manuscript.

Reviewer #1:

General Comments:

The homeboundness among elderly is a serious problem investigated in this study. The authors pointed the benefits of regular physical activity in different communities which is a good direction for public health priorities.

Our reply:

We really appreciate your constructive feedback. We have addressed each of your suggestions in the responses provided below. The page and line numbers shown correspond to the plain text with accepting Track Changes.

Comment 1:

My main concern refers to the literature cited by the Authors. 8 out of 22 (36%!) positions are self-citations. They are relevant to the article, as they are mainly the papers already based on the same study (JAGES), however I would like the Authors to find more international literature to refer to. Worth noticing fact is that in the discussion section the Authors refer to only 5 papers published by other authors. I would encourage them to discuss more international literature, as the discussion section should contain more comparison and references to existing findings.

Our reply:

Following the reviewer’s suggestions, we have added 10 references to the Introduction and Discussion sections [5–8, 24, 27–30, 32] and the corresponding sentences as necessary as follows (P9, L241–252; L276–279):

The strategies for enhancing the stages are mostly individual-oriented approaches, such as raising an individual’s consciousness and enhancing self-efficacy [28–30] … Another review has concluded that the transtheoretical model is a useful and suitable behavior model in creating, developing, and evaluating interventions to acquire and improve physical activity habits in older individuals [29].

Fourth, we could not consider the geographic characteristics of the communities which could contribute to physical activity [32]. This might be one of the unmeasured confounders and diminish the relationships identified in this study.

Comment 2:

I would also advise to simplify the title of the paper as it can be confusing for the readers.

Our reply:

We tried to make a simple title and removed the low-priority phrase: "from the JAGES".

We would like to thank the Reviewer for their helpful comments and hope that the revised manuscript is acceptable for publication in International Journal of Environmental Research and Public Health.

Reviewer 2 Report

We congratulate the authors of this article for the Japan Gerontological Evaluation Study (JAGES) project and the entire publication associated with it.
In this article, with a methodology very similar to previous articles such as Tsuji, T., Kanamori, S., Miyaguni, Y., Hanazato, M., & Kondo, K. (2019). Community-Level Sports Group Participation and the Risk of Cognitive Impairment. Medicine and science in sports and exercise, 51(11), 2217–2223. https://doi.org/10.1249/MSS.0000000000002050; the authors / researchers explore a new dimension.

We believe that the article and the results are of interest to readers and researcher in the field of the older people, physical exercise and social contagion.

Somes sugestions/apreciations:
Keywords: contextual effect and housebound are not Mesh term. We leave it to the authors to consider.

Introduction: Suggestion: Consubstantiate the introduction with other international references, between lines 35 to 48. The research carried out by this team is of great importance, but additional other references would be important.

Method
It seem unclear how the participants who exercise alone are classified. were added to those who do group exercises? Line 81 – “They were considered analytic participants for generating community-level variables of sports group participation … ” Who dooes individual exercise do to belongs to the group of non-participants?
Suggestion - make this information clearer.

Tabela 2 – sugestion: caption – Max: maximum; Min:minimum
Line 215 – To put [] in reference 10

Conclusion:
we consider that it would be interesting the authors make some considerations of this knowledge for society / for the political dimension of the municipalities and future investigations.

Author Response

Thank you for your time to review our manuscript and providing constructive feedback. We think that our manuscript is improved by addressing your feedback. Thank you very much.

We highlighted our changes using the Track Changes function in the revised manuscript.

Reviewer #2:

General Comments:

We congratulate the authors of this article for the Japan Gerontological Evaluation Study (JAGES) project and the entire publication associated with it.

In this article, with a methodology very similar to previous articles such as Tsuji, T., Kanamori, S., Miyaguni, Y., Hanazato, M., & Kondo, K. (2019). Community-Level Sports Group Participation and the Risk of Cognitive Impairment. Medicine and science in sports and exercise, 51(11), 2217–2223. https://doi.org/10.1249/MSS.0000000000002050; the authors / researchers explore a new dimension.

We believe that the article and the results are of interest to readers and researcher in the field of the older people, physical exercise and social contagion.

Our reply:

The authors really appreciate your thorough consideration of our manuscript and positive feedback. We have addressed each of your suggestions in the responses provided below. The page and line numbers shown correspond to the plain text with accepting Track Changes.

Comment 1:

Keywords: contextual effect and housebound are not Mesh term. We leave it to the authors to consider.

Our reply:

We decided that they were important terms in this study and left both of them.

Comment 2:

Introduction: Suggestion: Consubstantiate the introduction with other international references, between lines 35 to 48. The research carried out by this team is of great importance, but additional other references would be important.

Our reply:

Following the reviewer’s suggestions, we have added four references [5–8] to the relevant part (P1, L36–41).

Comment 3:

Method

It seem unclear how the participants who exercise alone are classified. were added to those who do group exercises? Line 81 – “They were considered analytic participants for generating community-level variables of sports group participation … ” Who does individual exercise do to belongs to the group of non-participants?

Suggestion - make this information clearer.

Our reply:

Regardless of the status of individual exercise, we focused on whether older individuals participated in a sports group or not. In other words, those who did only individual exercises were categorized as non-participants in a sports group. To make it clearer, we have revised the text as follows (P2, L84–87):

In the procedure of generating community-level variables of sports group participation, respondents who did not answer the extent of sports group participation were considered as "non-participants" in a sports group regardless of the status of individual exercise …

Comment 4:

Table 2 – sugestion: caption – Max: maximum; Min:minimum

Line 215 – To put [] in reference 10

Our reply:

Thank you for finding the mistakes. We corrected them (Table 2; P9, L241).

Comment 5:

Conclusion: we consider that it would be interesting the authors make some considerations of this knowledge for society / for the political dimension of the municipalities and future investigations.

Our reply:

Following the reviewer’s suggestions, we have added a suggestion for policy making perspectives as follows (P10, L288, 289):

A policy to increase sports groups, giving priority to communities where many older individuals with health problems live, may be effective.

We would like to thank the Reviewer for their helpful comments and hope that the revised manuscript is acceptable for publication in International Journal of Environmental Research and Public Health.

Reviewer 3 Report

Dear Editor,

I carefully read the article by Tsuji et al., which reports overall interesting results. Statistical analysis is appropriate and well performed.
I have only some minor comments for the authors:

  • Lines 36-38 - I suggest authors to move references 1-3 at the end of the sentence (before the period).
  • Line 80 - Authors should report age as mean plus/minus standard deviation.
  • Figure 1 and Lines 72-81 should be moved from "Methods" to "Results", as they regard the process of subjects' selection. In the methods, authors should only report inclusion and exclusion criteria of the study.
  • Lines 102 and 103 - Authors wrote "The choices for the answers were as follows: ≥4 days/week, 2–3 days/week, 1 day/week, 1–3 days/month, a few times/year, or zero". However, authors should add a reference to justify this classification.
  • English language needs to be carefully revised by a native English speaking person. In particular, verb tenses need to be carefully reviewed (in some passages, authors wrongly pass from present tense to past).
  • References are few in numbers and in most cases obsolete. In particular, authors should consider to replace ref. 16, 17 and 21 with newest articles.

Author Response

Thank you for your time to review our manuscript and providing constructive feedback. We think that our manuscript is improved by addressing your feedback. Thank you very much.

We highlighted our changes using the Track Changes function in the revised manuscript.

Reviewer #3:

General Comments:

I carefully read the article by Tsuji et al., which reports overall interesting results. Statistical analysis is appropriate and well performed. I have only some minor comments for the authors.

Our reply:

The authors really appreciate your thorough consideration of our manuscript and positive feedback. We have addressed each of your suggestions in the responses provided below. The page and line numbers shown correspond to the plain text with accepting Track Changes.

Comment 1:

Lines 36-38 - I suggest authors to move references 1-3 at the end of the sentence (before the period).

Line 80 - Authors should report age as mean plus/minus standard deviation.

Our reply:

Following the reviewer’s suggestions, we have revised (P1, L36; P4, L171).

Comment 2:

Figure 1 and Lines 72-81 should be moved from "Methods" to "Results", as they regard the process of subjects' selection. In the methods, authors should only report inclusion and exclusion criteria of the study.

Our reply:

Following the reviewer’s suggestions, we have revised the relevant part in the Methods and Results suctions (P2, L79–91; P4, L167–181).

Comment 3:

Lines 102 and 103 - Authors wrote "The choices for the answers were as follows: ≥4 days/week, 2–3 days/week, 1 day/week, 1–3 days/month, a few times/year, or zero". However, authors should add a reference to justify this classification.

Our reply:

Although the choices were set for convenience, we believe that there might be a little problem when determining whether or not they meet the validated criteria "going out 1 day/week or less" [13,18].

Comment 4:

English language needs to be carefully revised by a native English speaking person. In particular, verb tenses need to be carefully reviewed (in some passages, authors wrongly pass from present tense to past).

Our reply:

We had native speakers of English proofread our English writing throughout the entire manuscript.

Comment 5:

References are few in numbers and in most cases obsolete. In particular, authors should consider to replace ref. 16, 17 and 21 with newest articles.

Our reply:

Following the reviewer’s suggestions, we have added 10 references to the Introduction and Discussion sections [5–8, 24, 27–30, 32]. Certainly, we cited several articles that are not the latest; however, we think that they are representative literature that proposed the original concepts at an early stage. Therefore, while keeping them, we added recent studies.

We would like to thank the Reviewer for their helpful comments and hope that the revised manuscript is acceptable for publication in International Journal of Environmental Research and Public Health.

Reviewer 4 Report

The manuscript identifies an interesting issue concerning exercise in elderly people. The authors made a good effort to analyze several variables like the impact of the community, but others are lacking. Three main issues are important:

  1. Partially described in the supplementary file but not fully discussed. Most of the elders do not leave alone so there may be a role of the partner or family stimulating the exercise pattern, how do the authors could account for this issue?
  2. What are the most prevalent comorbidities of the group studied?
  3. And finally, the health issues are more prevalent in less educated people? 

Reviewer 5 Report

The authors demonstrated that older individuals living in a community area with a high prevalence of sports group participation were less likely to be homebound and have higher predisposition for exercise than those living in an area with low prevalence of participation. This reperesent an interesting point of interest.

Due to this, promoting sports groups in a community may be an effective strategy for increasing the enhancing interest and increasing the frequency of leaving the house even for non-participants in a sports group.

However some aspects in the manuscript need to be improved.

In my opinion the discussion needs to be upgraded by taking into consideration and expanding three main aspects beyond those already mentioned by the authors.

1) The authors should better comment in the discussion section on the difference between the 65-69 group (34.4% of participants) and the > 80 group (17.8%). People in these two groups can have very large differences in motility and this can be reflected in their predisposition to movement.

2) The authors should better comment on the geographical difference of origin of the subjects in the discussion section. The study aims to evaluate the population of Japan in general, but there are substantial differences in the habits of people in the different regions (for example Tokyo vs Hokkaido or Kyushu). What percentage comes from the different regions?

3) The authors should mention the fact that some people can be stimulated by a generally active society to take initiative to move and have social relations. However, other people, especially those who are elderly (> 80) or with disabilities, may become more depressed by being unable to participate in social activities. Please comment on this.

Round 2

Reviewer 4 Report

The authors have responded to all the queries.